# Aromas: Lovely to Smell and Nice Solvents for Polyphenols? Curcumin Solubilisation Power of Fragrances and Flavours [note 1]

**DOI:** 10.3390/molecules29020294

**Published:** 2024-01-05

**Authors:** Michael Schmidt, Verena Huber, Didier Touraud, Werner Kunz

**Affiliations:** 1Institute of Materials Resource Management, University of Augsburg, Am Technologiezentrum 8, D-86159 Augsburg, Germany; 2Institute of Physical and Theoretical Chemistry, University of Regensburg, D-93040 Regensburg, Germany; verena1.huber@chemie.uni-regensburg.de (V.H.); didier.touraud@chemie.uni-regensburg.de (D.T.)

**Keywords:** curcumin, aromas, green solvents, solubilisation, lactones, fragrances, flavours, polyphenols

## Abstract

Natural aromas like cinnamaldehyde are suitable solvents to extract curcuminoids, the active ingredients found in the rhizomes of *Curcuma longa* L. In a pursuit to find other nature-based solvents, capable of solving curcumin, forty fragrances and flavours were investigated in terms of their solubilisation power. Aroma compounds were selected according to their molecular structure and functional groups. Their capabilities of solving curcumin were examined by UV–Vis spectroscopy and COSMO-RS calculations. The trends of these calculations were in accordance with the experimental solubilisation trend of the solubility screening and a list with the respective curcumin concentrations is given; σ-profiles and Gibbs free energy were considered to further investigate the solubilisation process of curcumin, which was found to be based on hydrogen bonding. High curcumin solubility was achieved in the presence of solvent (mixtures) with high hydrogen-bond-acceptor and low hydrogen-bond-donor abilities, like γ- and δ-lactones. The special case of DMSO was also examined, as the highest curcumin solubility was observed with it. Possible specific interactions of selected aroma compounds (citral and δ-hexalactone) with curcumin were investigated via ^1^H NMR and NOESY experiments. The tested flavours and fragrances were evaluated regarding their potential as green alternative solvents.

## 1. Introduction

Polyphenols and flavonoids are of special interest due to their antioxidative, anti-inflammatory, antimicrobial, and anticancer properties [1,2,3,4]. Especially since the general awareness of health and environmental impacts has been increasing, products containing polyphenols are on the rise. Many flavour and fragrance compounds share a fundamental phenolic structure, like vanillin or curcumin. They are found in the corresponding spices grown in tropical climates. Some aromas like cinnamaldehyde and anisaldehyde show health-benefiting properties like antioxidant or antibacterial activity [5,6,7]. Additionally, most of the investigated aroma compounds like citral, limonene, and R-carvone, or lactones are found in plants, fruits, and spices [8]. Fragrances and polyphenols are often used together in perfumes or cosmetics. For example, in perfumery, natural α-tocopherol was found to better protect benzaldehyde, an easily oxidizable aroma component, from oxidation than butylated hydroxytoluene [9]. Another example is a study by Marteua et al. who examined the antioxidative power of polyphenols added to olfactory compounds in perfumes and essential oils [10].

In a recent paper [11], it was demonstrated that the extent of the solubilisation of curcumin in cinnamon-bark essential oil is strongly dependent on the cinnamon species and the nature and amount of molecules present in the essential oil. It was found that the solubilisation is linked to specific interactions between cinnamaldehyde and curcumin.

In continuation of Huber et al. [11], more aroma compounds were investigated regarding their capacity to dissolve curcumin. The solubility of curcumin is a research topic of interest. However, in these studies, curcumin is solubilised with surfactants [12], proteins [13], or nanocarriers [14], or it is complexed with cyclodextrins [15]. Here curcumin is often solubilised with organic solvents like ethanol, acetone, or DMSO prior creation of colloidal delivery systems [16]. This study provides a new approach to directly dissolve curcumin with aroma compounds, which was motivated by the simultaneous presence of polyphenols, fragrances, and flavours in natural products, food, beverages, perfumes, and cosmetic products. With this approach, extraction mixtures could be directly used in formulation, e.g., in the use of smart packing films similar to Jamroz et al. [17]. After a first test of commonly used aroma compounds like vanillin, citral, cinnamaldehyde, anethole, eugenol, and limonene, aroma compounds were chosen according to their functional groups and chemical structure (aldehydes, ketones, lactones, alcohols, or terpenoids), and their influence on the solubility of curcumin was examined. Apart from curcumin solubility in pure aroma compounds, binary mixtures with ethanol were tested, as ethanol is a commonly used solvent in the food and perfume industry and can also be derived naturally [18,19]. With the variety of molecules and their different functional groups, like alcohols, esters, aldehydes, and ketones, the underlying interactions between curcumin and solvent systems were investigated via COSMO-RS calculations and NMR measurements, providing a better understanding of the principles governing the curcumin solubilisation process.

## 2. Results and Discussion

Continuing the studies of Huber et al. [11], where aromas improved the solubility of curcumin even more than previously tested additives like triacetin and NADES [20,21,22], more aroma compounds were examined. Aroma compounds were compared depending on their functional groups by optical density measurements and COSMO-RS calculations. In the second part, the underlying solubility mechanisms between the aroma compounds and curcumin were instigated.

### 2.1. Solubility Screening

In the screening experiments, compounds with aromatic structures, monoterpenoids, cyclic ketones, lactones, and commonly used solvents were examined. Aromatic structure refers to the chemical definition of a molecule with cyclic structures and a conjugated π-electron system. All tested compounds with their chemical structure, c(x)–c(ethanol) ratio, and experimental and calculated concentrations are shown in Table 1. The compounds are listed with their increasing capacity to dissolve curcumin, which is represented by the multiplying factor c(x)–c(ethanol), with c(x) being the curcumin concentration in solvent x and c(ethanol) the concentration of curcumin in pure ethanol. Table 1 also contains the calculated chemical potential µ(solv) and the molar solubility log10(S) of the COSMO-RS calculations. For comparison to the calculated solubility, the decadic logarithm of the concentration log10(c) is given as well.

Based on this solubility screening, the following trend is established, where the highest curcumin solubility can be obtained with the addition of DMSO and the lowest with nonfunctionalized compounds or alcohols.

DMSO > δ-lactones > cyclic ketones/γ-lactones > aldehydes in conjugation to aryls > conjugated aldehydes and terpenoids with a carbonyl group > esters > ethers > nonfunctionalized compounds/alcohols.

The calculated chemical potential µ(solv) of curcumin vs. the experimentally determined concentration of curcumin in pure liquid solvents is shown in Figure 1. Figure 1b shows the section between 0.01 and 1 mol/L of the whole plot in Figure 1a. The datapoints are labelled according to the numbers in Table 1, and a confidence and prediction band are given. Generally, the curcumin concentration increases with the decreasing chemical potential.

Plotting the chemical potential against the logarithmic concentration gives a rough linear correlation. Points located above the linear fit correspond to an underestimated curcumin solubility, and for points below the straight, the curcumin solubility was overestimated. The same trend is observed when the logarithmic concentration is plotted against the predicted solubility log10(S) (cf. Appendix A). There are outliers of the confidence band and even two exceeding the prediction band, namely ethanol (1) and limonene (2). The other points that exceed the confidence band but not the prediction band can be assigned to different compound classes. Points (3)–(7) are linear compounds with a hydroxy function. Above the straight, trans-Anethole (8), an aromatic compound with a methoxy group and an allyl group in the paraposition, is located. Points (11)–(14) above the straight are ionones, terpenoids with a carbonyl group. Close to the ionones, carvone (23) and delta-tetradecalactone (19) are found, also containing a carbonyl group or a cyclic ester in the case of the lactones. At even higher concentrations, more lactones (24, 26, 27, and 32) are located above the linear fit. These lactones all have long alkyl chains. All these outliers either contain a hydroxy group or an unpolar part from alkyl or aromatic groups. Cyclohexanone (35), on the other hand, is a cyclic ketone that is located above the straight, while acetone (17), delta-valerolactone (41), and DMSO (42) are found below the straight of the linear fit. These four compounds exhibit strong polarity due to their functional groups (carbonyl, lactone, and sulfoxide) and, in contrast to the other outliers, are solvents with a high capacity to dissolve curcumin. The influence of different functional groups is discussed further in Section 2.1.1, Section 2.1.2, Section 2.1.3 and Section 2.1.4.

The capacity to dissolve curcumin was also investigated in binary ethanolic mixtures of liquid compounds (excludes vanillin and veratraldehyde). An overview of the curcumin solubility in ethanolic mixtures is compiled in heatmaps with the decadic logarithm of the concentration (a) or the calculated chemical potential (b), cf. Figure 2. The compounds are sorted according to the increasing curcumin concentration. The chemical potential is plotted in inverse order, as the solubility increases with decreasing chemical potential.

For the experimental solubility, a few trends are observed. First, the highest solubility is achieved by pure DMSO on the top right (cf. Figure 2a), while (*R*)-(+)-limonene results in the lowest observed solubility. Second, for the most part, the solubility in pure ethanol (=0% additive content in ethanol) is lower than in mixtures or pure solvents. Only for the last six entries (nerol to (*R*)-(+)-limonene) did the curcumin solubility decrease with an increasing addition of the solvent. Third, synergistic effects are observed between 40 and 80 mol% additive contents, meaning the curcumin solubility is higher in binary mixtures of solvent–ethanol than in pure ethanol or pure solvent. For the predicted solubilities, three trends are observed as well. Analogously to the experimental concentration, DMSO exhibits the highest and (*R*)-(+)-limonene the lowest curcumin solubility. Second, for the most part, the chemical potential in pure ethanol (=0% additive content in ethanol) is equal to or even higher than in mixtures or pure solvents. Third, a large part of the synergistic effects are observed between 20 and 80 mol% additive content. In comparison to heatmap (a), these effects are less distinct mostly due to the higher solubility of curcumin in ethanol. Overall, the differences between different additives are less pronounced for the predicted solubilities in Figure 2b. The total solubility trend coincides within both heatmaps, where the curcumin solubility increases from bottom to top and is improved by synergistic effects in binary solvent mixtures with ethanol. However, some discrepancies are still observed. The solubility in pure ethanol and at low additive contents is mostly lower than in the residual parts for the experimentally determined solubilities, while it is mediocre for the predicted solubilities. Second, the curcumin solubility in ethanolic mixtures ranging from citral to whisky lactone is overestimated while the solubility of carvone to massoia lactone in ethanolic mixtures is underestimated by COSMO-RS. These discrepancies probably are a result of intermolecular interactions. Nevertheless, the results of the COSMO-RS calculations provide a good estimation of the solubility trend.

Different functional groups strongly influence the solubility of curcumin. Hydroxy and nonfunctionalized groups affect the solubility of curcumin negatively, while methoxy, carbonyl (aldehydes or ketones), lactones, sulfones, sulfoxides, and aromatic groups exhibit a positive effect. The differences between them are discussed in the following parts (Section 2.1.1, Section 2.1.2, Section 2.1.3 and Section 2.1.4).

#### 2.1.1. Aromatic Compounds

All four aromatic compounds in Figure 3 share the base structure of benzaldehyde. The use of solid compounds like veratraldehyde and vanillin is limited by their solubility in ethanol, 30 mol% and 17 mol% respectively. The curcumin solubility is positively affected by an aldehyde in conjugation to the aryl group. The addition of one or two methoxy groups, p-anisaldehyde and veratraldehyde, further increases the solubility of curcumin. In contrast, the addition of a hydroxy group (vanillin) has a negative effect on the solubility of curcumin. It even negates the positive effect of the methoxy group as the solubility of curcumin with vanillin is even worse than with benzaldehyde, which does not have any additional functional group.

#### 2.1.2. Monoterpenoids

Next, two terpenoid classes are considered. The first group is derived from citral (cf. Figure 4a), which contains a formyl group in conjugation to a C-C double bond. Citronellal contains a formyl group without conjugation to a C-C double bond, which hardly affected the curcumin solubility in comparison to pure ethanol (grey line). Changing the formyl group to a hydroxy group without conjugation even decreased the curcumin solubility in the case of citronellol. The lack of a C-C double bond in the α-β position allows for free rotation of C-C bonds, which is sterically unfavourable. Additionally, intermolecular interactions have to be considered. The difference between aldehydes, citral, and citronellal is a result of the conjugation of the formyl groups and the formation of hemiacetals in ethanol. As observed with trans-cinnamaldehyde and hydrocinnamaldehyde, the C-C double bond in conjugation with the aldehyde reduced the amount of hemiacetal, which is formed [11]. The formed hemiacetal with its hydroxy group again is unfavourable for the solubilisation of curcumin.

The second terpenoid class considered are ionones, cf. Figure 4b. The difference between the ionones arises from the position of the C-C double bond, which is in conjugation with the carbonyl group in the case of β-ionone and pseudoionone. Apart from a possible hemiacetal formation, the interrupted conjugation of the π-electron system also results in a sterically unfavourable orientation, where the ring is perpendicular to the chain in the case of α-ionone. While pseudoionone is not a cyclic terpene, the conjugated electron system results in a linear orientation of the chain with little C-C rotation, further indicating the importance of a conjugated π-electron system.

#### 2.1.3. Lactones

For γ- and δ-lactones with different lengths in their side chain, a linear trend is observed, where the curcumin solubility decreased with the increased number of carbons in the chain, cf. Figure 5. This is due to the entropy of the alkyl chain, where the number of degrees of freedom increases with an increasing number of C-C single bonds.

Optical density measurements of the lactones shown in Figure 5 and two other lactones can be found in Appendix A. Whisky lactone with an additional methyl group was investigated in the case of γ-lactones, and massoia lactone with an additional double was examined for the δ-lactones. A comparison of the respective lactone, γ-octalactone, showed that the additional methyl group of whisky lactone was not favourable, probably due to steric effects. Comparing massoia lactone to δ-decalactone showed the same result, where the additional double bond was also not favourable. Unlike with whisky lactone, steric effects cannot be the reason, as the double bond hardly affects the orientation of the side chain and the previously discussed electronic effects would have suggested an increase with an increasing electron density at the functional group. Considering the assumption that the solubility of curcumin is also governed by hydrogen bonding, it is suspected that, due to the more dispersed π-electron system, the hydrogen-bond-acceptor ability of the lactone is affected negatively. 

#### 2.1.4. Solvents

Considering common solvents, a clear trend is observed, cf. Figure 6. DMSO with its sulfoxide group increased curcumin solubility the strongest. It is followed by cyclopentanone, a cyclic ketone, and sulfolane with a sulfone group. Acetone also improves the solubility of curcumin, but in comparison to cyclopentanone, it performs five times worse. Ethanol and 1-octanol as alcohols exhibit the lowest curcumin solubility of these solvents. Octanol, with a medium-length alkyl chain, performs even worse than ethanol due to the longer alkyl chain.

Considering these findings, hydrogen bonding seems to be governing the solubility capacity, where hydrogen-bond-donor groups affect the curcumin solubility negatively, while hydrogen-bond-acceptor groups generally improve the solubility. Intermolecular interactions with a focus on hydrogen bonding are discussed next.

### 2.2. Interactions

Based on the observed trend and the results of the COSMO-RS calculations, it was suspected that hydrogen bonding between curcumin and aroma compounds or solvents might govern the solubility [23]. First, sigma profiles and the Gibbs free energy are considered. In the second part, the presence of specific interactions, as found by Huber et al. [11], is investigated via NMR spectroscopy.

#### 2.2.1. Hydrogen Bonding

First, the σ-profile and σ-potential of the four used keto-enol curcumin conformers are considered, cf. Figure 7. The σ-profile is the probability distribution of the screening charge density (SCD) segments and is separated into three parts; σ < −0.01 e/Å^2^ represents the area of hydrogen-bond donors (HBDs), while σ > 0.01 e/Å^2^ is the area of hydrogen-bond acceptors (HBAs). Between −0.01 e/Å^2^ < σ < 0.01 e/Å^2^ is the nonpolar region. Due to the sign inversion from polarisation of the virtual conductor, the positively charged segments of HBDs appear at negative σ and the negatively charged segments of HBAs at positive σ. 

The σ-potential is a normalized distribution function of the chemical potential of a segment according to its screening charge density. A negative σ-potential below −0.01 e/Å^2^ represents an affinity for interactions with HBD, the molecule’s HBA capacity, and beyond 0.01 e/Å^2^, an affinity for interactions with HBA, the molecule’s HBD capacity. In contrast, the positive σ-potentials describe a lack of these interactions. Both provide insight into intra- and intermolecular interactions.

Looking at the σ-profile of curcumin (cf. Figure 7a), the nonpolar region is predominant, with a maximum at 0.005 e/Å^2^. It also exhibits peaks in the HBD and HBA regions. This is also represented by the σ-potential of curcumin (cf. Figure 7b), which shows affinity for HBAs and HBDs, due to its hydroxy, carbonyl, and methoxy functionalities. In the nonpolar region, curcumin has a µ(σ) > 0, reflecting unfavourable interactions with nonpolar surfaces. Its hydroxy groups at the benzyl rings and the keto-enol moiety are HBDs, whereas its methoxy groups and the carbonyl group of the keto-enol moiety curcumin can act as an HBA. As curcumin has a symmetric σ-profile, intramolecular hydrogen bonding can occur. In the solubility screening, compounds containing HBDs were found to reduce the capacity to dissolve curcumin, while the presence of HBAs improved the curcumin solubility. Looking at the contributing groups of curcumin for HBA and HBD, it becomes evident why this might be the case. The HBD contribution is made up of three hydroxy groups, while the HBA contribution consists of two methoxy groups and only one carbonyl group. These groups differ; the lone pairs of the carbonyl are in sp2 orbitals, while the lone pairs of the methoxy group (ether) are in sp3 orbitals. While only a slight difference in hydrogen-bond strengths between ethers and carbonyls is reported, an orientation preference of the lone pair exists for carbonyls but not for ethers [24]. Thus, the HBD contribution is probably not completely compensated for by the present methoxy groups, resulting in a small electrostatic misfit which is only compensated for upon the addition of external HBAs, like carbonyls, esters, or lactones. A variety of σ-profiles of tested compounds with different functional groups are shown in Figure 8.

All compounds have one or more broad peaks within the nonpolar region and peaks beyond 0.01 e/Å^2^. Carbonyls and lactones have a peak at around 0.013 e/Å^2^. However, the lactones’ p(σ) are larger than the carbonyls’. As for carbonyls, slight differences of the p(σ) in the HBA region are observed, but there are considerable differences in the nonpolar region, e.g., trans-cinnamaldehyde and citral (cf. Figure 8b). The nonpolar region of trans-cinnamaldehyde is a similar shape to that of curcumin, while citral only has a peak at 0.00 e/Å^2^ coming from the terpenoid structure. Van der Waals interactions in the nonpolar region can occur; however, they compete with the entropy of the alkyl chain, as can be seen with delta-lactone (cf. Figure 8e). For p-anisaldehyde, the addition of a methoxy group broadens the HBA peak, increasing its HBA abilities, whereas a methoxy group alone (trans-anethole) results only in a weak peak at 0.011 e/Å^2^ (cf. Figure 8b).

Sulfolane and DMSO (cf. Figure 8f) have considerably different σ-profiles than the other compounds. The peak of sulfolane is also at 0.013 e/Å^2^ but its p(σ) is considerably higher than that of carbonyls and lactones due to the sulfone group. The exceptionally high curcumin solubility in DMSO can probably explain DMSO’s peak at 0.18 e/Å^2^, which is the highest surface charge density of all tested compounds. Due to its similarity to the HBA peak of curcumin, it is suspected that DMSO can specifically interact with the keto-enol moiety of curcumin. This is further discussed in the following and Section 2.2.2 with respective NMR measurements.

Ethanol (cf. Figure 8a) is the only compound that has positive and negative surface segments providing potentially hydrogen-bonding surface segments of opposite polarities. Thus, the addition of ethanol to the other compounds reduces the electrostatic misfit of them, which occurs due to their lack of surface segments on the left side between −0.01 e/Å^2^ and −0.02 e/Å^2^. Thus, binary ethanolic mixtures yield an excess of surface segments beyond 0.01 e/Å^2^, which complement the electrostatic misfit of curcumin resulting in higher curcumin solubilities in binary mixtures than in pure compounds.

Considering the reported values of HBA and HBD abilities in the literature [25,26,27,28], a relation between high curcumin solubilities and hydrogen-bond interactions is observed as well. The affinities correspond to observation with lactones, where delta-lactones are better than gamma-lactones, as they have higher HBA affinity [27]. Strangely, the hydrogen-bonding affinities do not correspond to cyclic ketones, where a six-membered ring should be better than a five-membered ring as well [27]. As the COSMO-RS calculations coincided with the experimental curves, it is suspected that the molecule structure affects the curcumin solubility. The hydrogen-bonding abilities of cyclic ketones are stronger than those of linear ketones, as well as corresponding to determined solubilities [27]. Also, it is reported that DMSO interacts differently regarding hydrogen bonding [27]. The special behaviour could be due to the additional hydrogen bonding that can occur with DMSO [29].

This proposition is also in accordance with the previous observation of Huber et al., where NADES and carboxylic acids could improve the solubility of curcumin in ethanol [22,30]. Both substance classes have HBA and HBD groups. Acids like pyruvic acid can increase curcumin solubility (3000 a.u.). However, the HBD groups restrict the solubility [22]. For NADES-containing solvent systems, absorbance values of up to 8000 a.u. were reported [30]. As HBD and HBA are essential for the formation of deep eutectic systems, the negative effect of HBD has probably less effect on the curcumin solubility due to the reduced electrostatic misfit. Most of the aroma compounds investigated in this study have only weak HBD groups if any at all. Thus, the positive influence of HBA on solubility dominates. Ethanolic aroma mixtures with, e.g., cinnamaldehyde, were roughly twice as good as NADES, with an optical density of up to 15,000 a.u. (cf. Figure 1). δ-lactones with an optical density of roughly 30,000 a.u. are again twice as effective as the aromatic flavour compounds. The highest solubility was obtained with DMSO (80,000 a.u.), cf. Figure 2. To further investigate this proposition, the energy differences between these interactions were considered.

The ΔG-values for a variety of additives with different functional groups are listed in Table 2. The obtained ΔG values are in the range of hydrogen bonding 2–12 kT [31], thus, indicating that HBA and HBD abilities are responsible for curcumin solubility.

As reported in the literature, the hydrogen-bonding network at the keto-enol moiety is an essential parameter for curcumin solubility and its physicochemical properties [32,33,34,35,36]. It readily interacts with HBAs and HBDs. Depending on the interaction, different enol conformers are obtained [35]. Only in the closed trans form is the planar hexagonal structure rigid; in all other cases, the symmetry and planar structure are broken, which enables rotation of aryl groups of curcumin, probably affecting the solubility.

Additionally, hydrogen bonds with the hydroxy and methoxy groups in the aromatic ring of curcumin are present [33]. HBAs and HBDs can also interact with these groups, which could be a reason for the large changes in the chemical shift in the aromatic hydroxy groups of curcumin with trans-cinnamaldehyde [11]. To prove this hypothesis, HBA–HBD interactions between curcumin and different additives should be investigated via IR and Raman measurements similar to procedures reported in the literature [23,29]. The system should also be studied via solvatochromism, which could provide more information in that regard. Nevertheless, these results provide a good baseline to select solvents/additives according to their functional groups.

#### 2.2.2. NMR Measurements

Interactions between aroma compounds and curcumin can be detected by changes in the chemical shift of curcumin protons in ^1^H-NMR [37,38] and intra- and intermolecular interactions can be observed via cross peaks in NOESY measurements [39,40]. First, curcumin in different deuterated solvents was investigated and the diketo–keto-enol ratio was determined. Then, curcumin in two citral or δ-hexalactone, methanol-d4 mixtures was compared to the previous study and put into the context of the solubility screening.

First, the solubility of curcumin (c.f. Figure 9) in deuterated solvents is evaluated. The equilibrium of the keto-enol tautomerism is dependent on the used solvent [41], as can be seen in Table 3. While roughly 10% of the diketo form is present in methanol-d4, only 1% is found in acetone-d6 and DMSO-d6. Looking at the curcumin concentration of the respective nondeuterated solvents (cf. Table 3), the dissolvable curcumin concentration increases about tenfold going from methanol-d4 to acetone-d6. In the case of DMSO, the curcumin solubility is a lot higher. DMSO seems to be a special solvent, one that can probably interact with curcumin via a different mechanism.

The hydrogen atoms (cf. Figure 9) of the hydroxy groups (3, 6, 22) and the hydrogen atoms between the keto-enol group (27) are not visible in methanol-d4 (cf. Appendix A) due to fast proton exchanges. In acetone-d6, only the hydroxy group (22) is not visible, as the signal is very weak and broad (cf. Appendix A). In DMSO-d6, all hydrogen atoms of curcumin are visible (cf. Appendix A). Short-distance intra- and intermolecular interactions can be observed with NOESY. Cross peaks between the hydrogen nuclei in curcumin can be seen in all three solvents (cf. Appendix A). In acetone-d6 and DMSO-d6, cross peaks between all proton signals are visible; while in methanol-d4, only cross peaks between neighbouring and nearby protons are visible. However, only in methanol-d4 and acetone-d6 cross peak signals with an inverse sign to the diagonal peaks are observed for all visible hydrogen nuclei. In DMSO-d6 all observed cross peaks have the same sign as the diagonal signals. This indicates a different magnetic environment, supporting the assumption that DMSO interacts differently with curcumin than, for example, with acetone.

Huber et al. [11] investigated the present interactions of curcumin with trans-cinnamaldehyde (shown for comparison in Table 4); here, interactions with one compound of each of the other two aroma classes (terpenoids or lactones) were chosen. Analogous to ethanol in the UV–Vis experiments, methanol-d4 was used for ^1^H and NOESY measurements, and the aroma compounds citral and δ-hexalactone were examined. Changes in the chemical shift of curcumin are observed, while no changes occur for the aroma compounds themselves (cf. Appendix A). An upfield shift of the chemical shift corresponds to an increase in electron density at the nucleus, while a downfield shift corresponds to a decrease in electron density at the nucleus. Analogously to trans-cinnamaldehyde, citral reacts with methanol-d4, and a hemiacetal is formed (cf. Appendix A). For citral, roughly 54% of the hemiacetal is observed.

In methanol-d4, no signals of the hydroxy groups (3&6)and (22) or the keto-enol proton (27) are visible in ^1^H NMR spectra, due to fast proton exchange with the deuterated solvent. The addition of trans-cinnamaldehyde results in pronounced changes in the chemical shifts, indicating an increasing electron density of the aromatic ring in the presence of cinnamaldehyde, while the electron density at the allyl protons (19&20) and (21&24) is decreased.

In a mixture citral–methanol-d4, a signal for proton (27) is observed. In contrast to trans-cinnamaldehyde, not all signals experience a change in chemical shift. Only the protons in the aromatic ring (11&12) experience a slight upfield shift (from 7.11 to 7.09 ppm) while the protons (7&16) experience a slight downfield shift (from 7.22 to 7.23 ppm) and the adjected allyl protons (19&20) as well (from 7.57 to 7.58 ppm). For the mixture of δ-hexalactone and methanol-d4, all curcumin protons are visible in the ^1^H NMR spectra, except for the hydroxy proton (22). Even though methanol poses a chance for fast proton exchange with hydroxy groups, a signal for proton (3&6) is observed. All other signals experience a slight downfield shift. Unlike with trans-cinnamaldehyde, no significant changes in the chemical shift are observed with citral and δ-hexalactone, indicating different underlying interactions with curcumin.

In the NOESY measurement with trans-cinnamaldehyde, only cross peaks with methoxy groups of curcumin (1&4) were reported [11]. The NOESY spectrum of citral contains a variety of cross peaks between curcumin and citral (cf. Appendix A). The methoxy groups of citral (5) and (10&11) and protons (7) especially interact with all visible curcumin protons. No cross peaks with the formyl group of citral are observed. In the mixture of δ-hexalactone–methanol-d4, nearly a complete set of cross peaks between curcumin and the lactone is observed (cf. Appendix A). Only the methoxy groups of curcumin (1&4) do not interact with proton (5) of δ-hexalactone. Unlike with cinnamaldehyde, no specific interactions between the two groups are observed with citral or δ-hexalactone not alike. This might also explain why predictions with COSMO-RS coincide better with terpenoids and lactones than with aromatic aroma compounds, apart from the formation of hemiacetal.

## 3. Evaluation

More effective solvents or solvent additives for the solubilisation of curcumin were found. Lastly, they will be evaluated regarding the principles of green chemistry [43]. For each group (aromatic, monoterpenoids, and lactones) a few compounds were chosen and evaluated regarding availability, production, toxicity, and industrial interest.

Of the aromatic compounds, trans-cinnamaldehyde and p-anisaldehyde performed the best. Cinnamaldehyde was already evaluated in a previous study and poses a potential alternative for curcumin extraction [11]. Anisaldehyde is commonly found in star anise [44] and is produced industrially via oxidation of, e.g., anethole [8]. Around 500–800 tons of star-anise essential oil is produced in China annually [45]. It is approved by the Flavour and Extract Manufacturers Association of the United States (FEMA) and the World Health Organization (WHO) to be used as a flavouring agent [46,47]. It is also used as an intermediate in many industrial processes [8]. Another well-performing aromatic additive is benzaldehyde. It is commonly found in bitter-almond oil and many other essential oils and is used as a fragrance and flavouring agent. It is also used as a starting material for the synthesis of many flavours and fragrances [8]. Benzaldehyde is regarded as generally safe by the WHO [47,48]. The annual production volume of synthetic benzaldehyde is over 7000 tons, and around 100 tons of natural benzaldehyde are produced each year [45]. Lately, more sustainable synthesis [49] and extraction routes are reported [50]. Eugenol is evaluated as well, as it is often found in combination with cinnamaldehyde in essential oils. It is present in cloves and obtained via extraction of the respective essential oils. Industrially, the synthetic routes are not important [8]. Due to its use in the production of vanillin, eugenol is an important aroma compound in industry [45]. It is used in perfumery and as a flavouring agent. However, due to possible cytotoxic and genotoxic effects [51], an acceptable daily intake (ADI) of 2.5 mg/kg body weight is given by FEMA [52]. Apart from classical use in perfumery or as a flavour, new applications for trans-cinnamaldehyde and eugenol arise in packaging films due to their antifungal and antibacterial properties [53,54]. This could be an interesting application for mixtures of these aromas with curcumin. The aromatic aroma compounds are generally regarded as safe and have high potential in the solubilisation process of curcumin and other polyphenols.

Citral is found naturally in fruits and herbs [55]. It is used in large quantities as an intermediate in vitamin A synthesis. For this application, it is synthesized from β-pinene or linalool, while extraction and isolation from essential oils are preferred for the perfume industry [55,56]. The annual production volume of essential oils high in citral (60–80%) is 1000 tons [45]. The production volume of synthetic citral is much larger, where BASF alone is able to produce 40 000 tons per year [57]. Its use as a fragrance and flavour agent is approved by the European Food Safety Authority (EFSA) and FEMA. However, due to its allergenic potential and irritating properties, an ADI of 0.5 mg/kg body weight is given by the WHO and EFSA [58,59]. Similar to trans-cinnamaldehyde and eugenol, the use of curcumin extracts with lemongrass essential oils, which are high in citral [60], in intelligent packaging films was reported recently [17]. Most synthetic citral is used for the synthesis of β-ionone in the production of vitamin A, but it is also used in perfumery and as a flavouring substance [45,55]. α- and β-ionones are also commonly found in flowers (violets) and fruits (apricots and berries) [55,61]. Between 4000 and 8000 tons of β-ionone are produced industrially each year [62]. The synthetic route uses citral and acetone as starting materials. In the first step, pseudoionene is formed, which is cyclized to α-ionone and then rearranged to β-ionone [45,56]. Lately, new biosynthetic routes are reported [63]. Applications in fragrance and flavour industries of both ionones and pseudoionone are approved by the WHO [64,65]. The last monoterpenoid considered is carvone, which occurs in caraway or spearmint oil with concentrations between 60 and 80% [55,66]. While carvone was obtained from hydrodistillation, nowadays carvones are prepared synthetically from the respective limonene (+) or (−) isomer. Carvones are used as flavouring agents in food, beverages, and cosmetic products [55]. They are also used as starting materials for natural substance synthesis [67]. Approximately 1200 tons are used annually [56]. Carvone is approved by FEMA. However, due to its allergenic potential, an ADI of 0.6 mg/kg body weight is given [68]. These monoterpenic aroma compounds are all found in plants and can be obtained from renewable sources [45]. Apart from their irritating and allergenic properties, they pose interesting natural solvent alternatives for curcumin and probably more polyphenols. Lastly, lactones are evaluated.

In nature, mostly saturated and unsaturated γ- and δ-lactones are found, as they are formed by intramolecular esterification of the corresponding hydroxy fatty acids. They occur in fatty foods and also in fruits, like peaches and plums [8,61]. Industrially, they are synthesized via the radical addition of primary fatty alcohols and acrylic acid. However, due to the high demand for natural γ- and δ-lactones in the aroma industry, biosynthetic processes are mainly used which also drastically decreased the price per kilogram and increased production volume to several tons per year [45]. The production of lactones via the biotransformation of waste materials or from natural sources was reported recently [69]. These lactones are commonly used in the food and beverage industry, as well as in perfumery due to their fruity and nutty/fatty aroma [8,45]. The characteristic flavours and fragrances like peach, plum, and coconut result from the different lengths of side chains and substituents [70]. Both lactone groups are approved by FEMA [71] and EFSA [72], with an exception for γ-butyrolactone and γ-valerolactone. Due to the fast metabolization of γ-butyrolactone to 4-hydroxybutanoic acid in the human body [73], it is only an industrial solvent. However, γ-valerolactone was recently re-evaluated regarding its toxicity and biodegradability and can be classified as a green solvent [74]. In contrast to larger lactones, γ-and δ-lactones do not have antioxidant or anti-inflammatory effects, which could be desirable properties in products [75]. By using lactones to dissolve curcumin, its antioxidant, antifungal, antibacterial, and antiviral properties could be introduced into products [76,77,78]. The two lactone groups present the highest potential as natural green solvents, due to an interest increase in nature-derived lactones over the last years, which greatly improved their availability. Additionally, the large variety of flavours and fragrances provides a nice palette for applications.

Most aroma compounds are generally regarded as safe. They are all approved as fragrance and flavouring agents, and thus, have high potential as new green-solvent alternatives for polyphenols.

## 4. Materials and Methods

### 4.1. Materials

All used chemicals are listed in Table 5. All chemicals were used without further purification.

### 4.2. Methods

#### 4.2.1. Solubility Screening

For the screening experiments, samples were prepared analogously to Huber et al. [11,22]. One g mixtures of an aroma compound or a solvent and ethanol with different weight ratios (0–100 wt%) were prepared. For solid aroma compounds like vanillin, its solubility in ethanol at room temperature was determined first, and weight ratios (aroma compound–ethanol) below to solubility maximum were selected. The homogenous solutions were supersaturated with synthetic curcumin and stirred for 1 h at room temperature. Afterwards, they were filtered with 0.45 μm PTFE syringe filters to remove the excess curcumin.

#### 4.2.2. Optical Density Measurements

The optical density was measured to qualitatively assess the solubility of curcumin in the different aroma–ethanol mixtures. The measurements were performed analogously to previous studies [11,20,21,22]. The measurements were performed via UV–Vis spectroscopy in the spectral range from 700 nm to 350 nm, using a Lambda 18 UV–Vis spectrometer by Perkin Elmer (Waltham, MA, USA). The samples were diluted accordingly with ethanol. Quartz glass cuvettes were used, and the measurement was performed at room temperature. The optical densities of different samples were compared at a wavelength of 425 nm.

#### 4.2.3. NMR Measurements

For promising binary mixtures found through optical density measurements, NMR spectra were recorded. This methodology was adopted from [11,22]. An aroma compound–methanol-d4 ratio of 30/70 (n–n) was used and 0.24 mmol of curcumin was added. To correctly assign NMR peaks and evaluate the change in chemical shift, reference spectra of curcumin in the deuterated solvents and of the aroma compounds (citral and δ-hexalactone) in the methanol-d4 were recorded. The prepared solutions were transferred into NMR tubes. The spectroscopy experiments were performed on an Avance III HD 400 MHz spectrometer equipped with a 5 mm BBO 400S1 BBF-H-D sample head with a Z-gradient at standard conditions by Bruker (Billerica, MA, USA). The samples were measured with the preprogrammed methods for ^1^H and NOESY, while the scan rate was increased for the NOESY spectra to achieve a better signal–noise ratio. All spectra were processed and analysed with Mestre Nova (version 14.1.2).

#### 4.2.4. COSMO-RS Theory

The conductor-like screening model for real solvents (COSMO-RS) theory is a powerful method to predict a variety of physicochemical properties by combining quantum chemistry with fast statistical thermodynamics. First, the screening charge density *σ* is calculated by embedding the molecular structure in a virtual conductor. The screening charge density of the molecule polarizes the ideal conductor and a cavity with inverse polarization to the molecules’ polarization is obtained (COSMO surface). COSMO-RS splits this cavity into smaller segments and utilises them for statistical thermodynamics, where it is assumed that all relevant molecular interactions consist of local pairwise interactions of these surface segments [79,80,81,82]. 

The chemical potential *µ^i^_s_* of a solute *i* in a solvent *s* can be calculated from the σ-potential *μ_s_*(*σ*) and σ-profiles *p*(*σ*) of a molecule (cf. Equation (1)). The σ-potential describes the affinity of the surface segments of the solvent *s* to the polarized surface, e.g., the capacity to interact with a hydrogen-bond acceptor. The σ-profiles are probability functions of the number of surface segments with a specific screening charge density of the system and reflect the polarity of a compound. The combinatorial term *µ^i^_c,s_* takes the size and shape differences of the molecules into account [82,83].
(1)$$\mu_{s}^{i}=\mu_{c,s}^{i}+\int p^{i}\left(\sigma \right)\mu_{s}\left(\sigma \right)d\sigma$$

#### 4.2.5. COSMO-RS Calculations

COSMOthermX (version 19.0.4) by COSMOlogic GmbH and Co. KG [84] was used to calculate the chemical potential of curcumin (keto-enol form) in pure solvents and binary solvent mixtures of different ratios (n–n). The calculations of the chemical potential at infinite dilution have been performed on the TZVPD-FINE level. Molecules were taken from the COSMObase TZVPD-FINE 19.0 database. For molecules that were not found in the database, conformer COSMO files were calculated with COSMOconfX (version 4.3) by COSMOlogic GmbH and Co. KG [85] on the TZVPD-FINE level. For the calculation of the keto-enol tautomer of curcumin, only the COSMO files of the four sensible conformations were used. The COSMO surfaces are shown in Appendix A.

## 5. Conclusions

The motive of this study was to qualify the use of aroma compounds as effective, natural solvents suitable for the solubilisation of curcumin, providing a platform for formulation in this field, and understanding the underlying solubility mechanism. A solubility screening of curcumin with over forty aroma compounds and classical solvents was conducted and analysed with UV–Vis spectroscopy and COSMO-RS. The HBA and HBD abilities of curcumin, aroma compounds, and solvents were evaluated based on their σ-profiles. The results were put into the context of typical bond interaction energies (ΔG). A compound of each aroma class—aromatic compounds, monoterpenoids, and lactones—was chosen for further analysis with ^1^H NMR and NOESY. Lastly, the aroma compounds were evaluated regarding their potential as alternative green solvents. Important results are compacted in the following bullet points.

Most tested aroma compounds and solvents exhibited better solubilisation power than ethanol. Aromatic aromas like trans-cinnamaldehyde, p-anisaldehyde, and benzaldehyde, as well as monoterpenoids like citral, carvone, and ionones, increased curcumin solubility 10 to 30 fold in comparison to ethanol. Even higher increases of up to 60 fold were achieved with γ- and δ-lactones. For these lactones, an inverse linear correlation between the increase in solubility and the length of the alkyl chain was found. The highest increase in curcumin solubility was achieved with DMSO (factor 160). For nearly all tested compounds, synergistic effects were observed in binary ethanolic mixtures. From the solubility screening, a reliable trend according to the structure and functional group of aroma compounds could be established, which is in accordance with the predicted solubilities via COSMO-RS.

Considering the σ-profiles and bond interaction energies, the solubility mechanism of curcumin was found to be based on hydrogen bonding. Aroma compounds and especially binary ethanolic mixtures of them exhibit an excess of HBA abilities, which complement the hydrogen-bonding abilities of curcumin, reducing the total electrostatic misfit. The highest curcumin solubility was achieved with DMSO. Its σ-profile differs considerably from all other tested compounds. In the subsequent NOESY measurements of curcumin in DMSO-d6, a change in the magnetic environment of curcumin was found with DMSO as well. Thus, it was assumed that DMSO interacts specifically with curcumin’s keto-enol moiety. This change was not observed for other solvents or aroma compounds. A few selected aroma compounds were also analysed via 1D and 2D NMR spectroscopy. Unlike with trans-cinnamaldehyde, no specific interactions were found with citral or δ-hexalactone in the NMR measurements.

All three aroma classes (aromatic compounds, monoterpenoids, and lactones) were evaluated regarding their potential as alternative green solvents. Due to the rising interest in aroma compounds, annual production volumes range from a few hundred to thousands of tons. As aroma compounds are found in nature, they also can be produced from renewable sources via biotransformation and biosynthetic processes. All aromas are approved for flavour and fragrance applications by the WHO.

Summarized in bullet points.

A list of curcumin solubility in over forty aroma compounds and solvents was presented;The solubility trend of the COSMO-RS calculations is in accordance with the experimental trend,DMSO > δ-lactones > cyclic ketones/γ-lactones > aldehydes in conjugation to aryls >conjugated aldehydes and terpenoids with a carbonyl group> esters > ethers > nonfunctionalized compounds/alcohols;Synergistic effects in binary ethanolic mixtures were observed for nearly all aroma compounds;Hydrogen bonding as the governing principle of curcumin solubilisation;Good solvent (-mixtures) exhibit an excess of HBA abilities;In contrast to trans-cinnamaldehyde, no specific interactions between curcumin and citral or δ-valerolactone were found;Production of aroma compounds increased consequently, decreasing their price;Most aroma compounds are generally regarded as safe;Suitable approach for new applications, e.g., packing films.

The investigated aroma compounds are viable green-solvent alternatives. Many new questions regarding the solubilisation of other hydrophobic polyphenols, extractions, and product formulations arise. Suitable aromas can be chosen to increase extraction yields, and extracts can be directly used in the formulation. 

We are not able to solve the mysteries of solubilisation with aromas in one article, but we hope that this study motivates further studies in this direction.

## Figures and Tables

**Figure 1 molecules-29-00294-f001:**
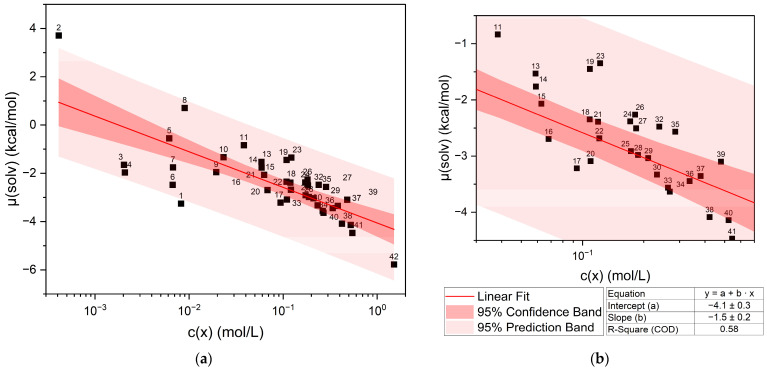
Chemical potential of curcumin µ(solv) vs. the logarithmic curcumin concentration c(x) in the respective liquid solvents and a linear fit of the solubility trend. (**a**) shows the whole plot, while (**b**) shows the section between 0.03 < log10(c(x)) < 0.7 for better identification of the datapoints. The numbering of labels refers to the list in Table 1. Vanillin and veratraldehyde are excluded due to being solid compounds.

**Figure 2 molecules-29-00294-f002:**
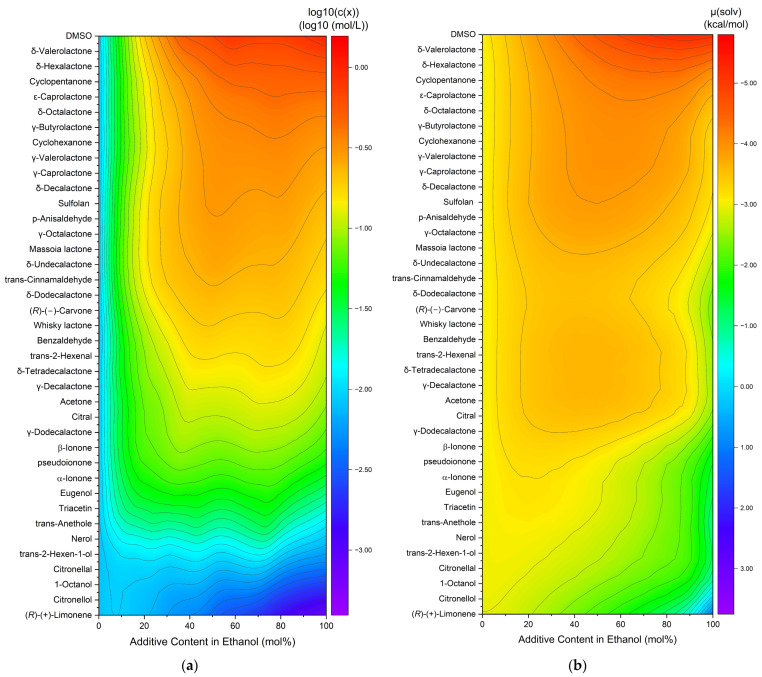
Heatmaps of curcumin solubility in binary ethanolic mixtures of the added liquid compounds. (**a**) Experimentally determined curcumin concentration in logarithmic plot and (**b**) with the calculated chemical potential of curcumin. Vanillin and veratraldehyde are excluded due to being solid compounds.

**Figure 3 molecules-29-00294-f003:**
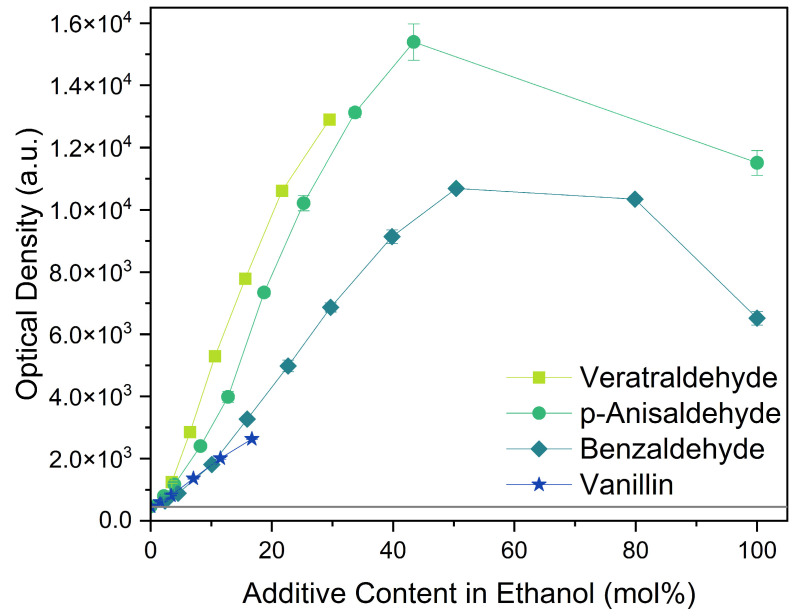
Optical densities in arbitrary units (a.u.) of curcumin in binary ethanolic mixtures with aromatic compounds. Veratraldehyde is represented by light-green squares, p-anisaldehyde by green circles, benzaldehyde by dark-green diamonds, and vanillin by blue stars. The solubility of curcumin in ethanol is represented by the grey horizontal line. The UV–Vis samples were analysed at a wavelength of 425 nm.

**Figure 4 molecules-29-00294-f004:**
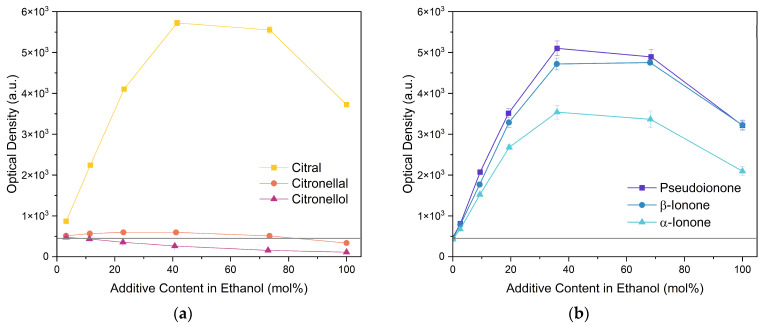
Optical densities (in arbitrary units, a.u.) of curcumin in binary ethanolic mixtures with monoterpenoids. (**a**) Citral derivates; citral is represented by yellow squares, citronellal by orange circles, citronellol by red triangles, and (**b**) ionones; pseudoionone is represented by purple squares, β-ionone by blue circles, and α-ionone by light blue triangles. The solubility of curcumin in ethanol is represented by the grey horizontal line. The UV–Vis samples were analysed at a wavelength of 425 nm.

**Figure 5 molecules-29-00294-f005:**
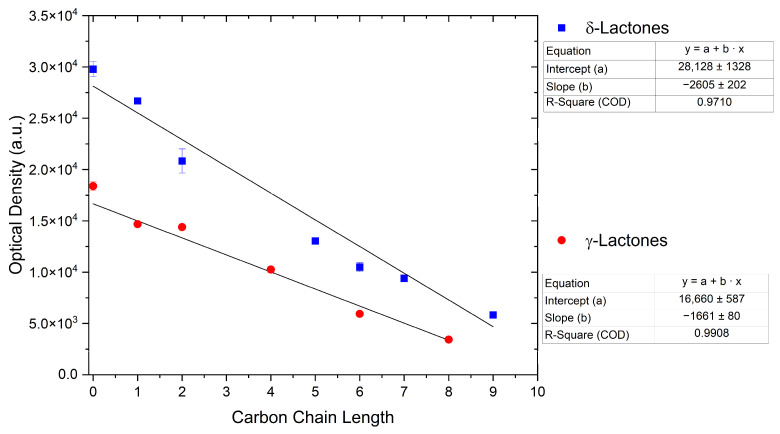
Optical density (in arbitrary units, a.u.) vs. number of carbons in the side chain of δ-lactones (blue squares) and of γ-lactones (red circles) with respective fit functions. The UV–Vis samples were analysed at a wavelength of 425 nm.

**Figure 6 molecules-29-00294-f006:**
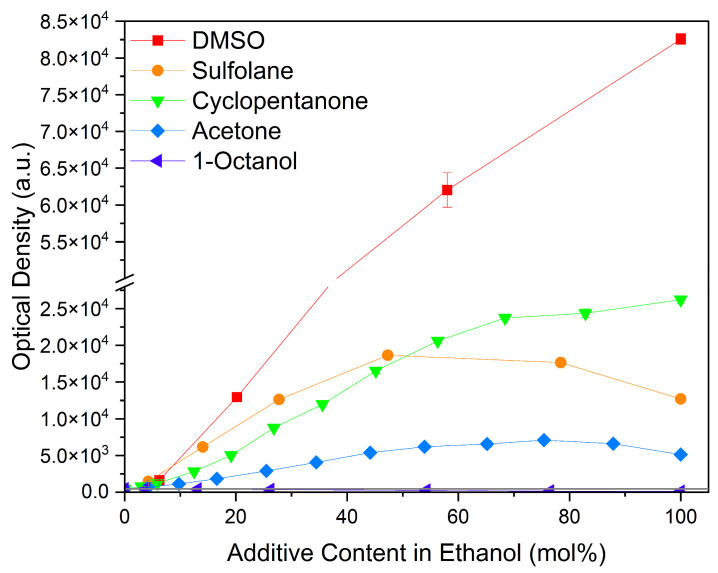
Optical densities in arbitrary units (a.u.) of curcumin in binary ethanolic mixtures with common solvents. DMSO is represented by red squares, sulfolane by orange circles, cyclopentanone by green down-facing triangles, acetone by blue diamonds, and 1-octanol by purple left-facing triangles. The solubility of curcumin in ethanol is represented by the grey horizontal line. The UV–Vis samples were analysed at a wavelength of 425 nm.

**Figure 7 molecules-29-00294-f007:**
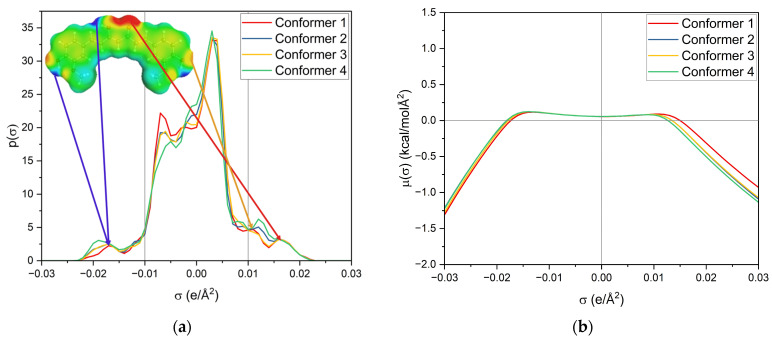
(**a**) σ-profile and (**b**) σ-potential of the four curcumin keto-enol conformers. The structures and σ-surfaces of these conformers are shown in Appendix A.

**Figure 8 molecules-29-00294-f008:**
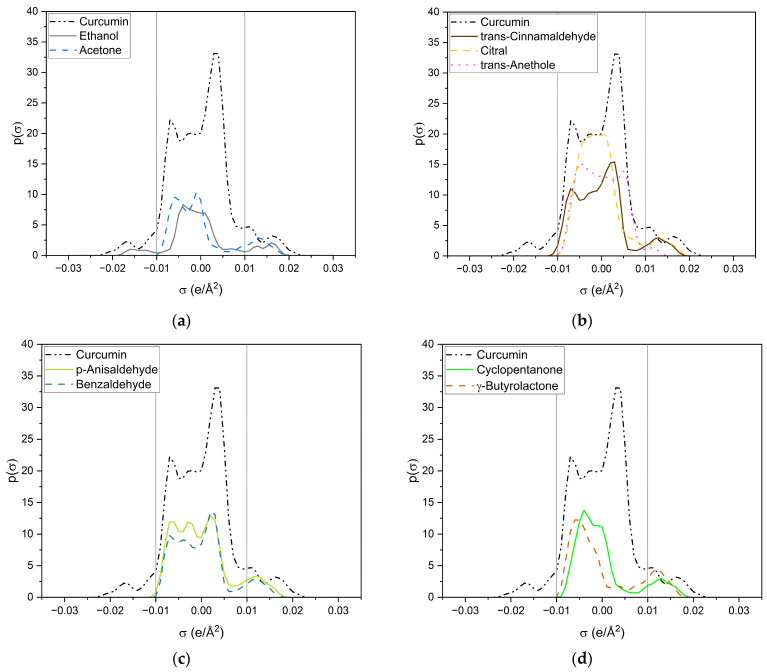
σ-profiles of curcumin (black dashed-dotted-dotted line) and a variety of investigated compounds. (**a**) Ethanol is represented by a grey solid line and acetone by a blue dashed line; (**b**) trans-cinnamaldehyde is represented by a brown solid line, citral by a yellow dashed line and trans-anethole by a pink dotted line; (**c**) p-anisaldehyde is represented by a light-green solid line and benzaldehyde by a dark-green dashed line; (**d**) cyclopentanone is represented by a green solid line and γ-butyrolactone by an ochre dashed line; (**e**) δ-valerolactone is represented by a dark-turquoise solid line and δ-decalactone by a cyan dashed line; (**f**) DMSO is represented by a red solid line and sulfolane by an orange dashed line.

**Figure 9 molecules-29-00294-f009:**
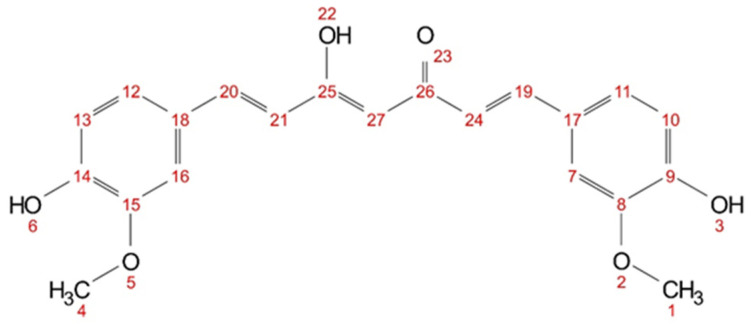
Chemical structure of curcumin (keto-enol) with numbered atom positions (1–27).

**Table 1 molecules-29-00294-t001:** List of investigated compounds with their chemical structure, change in solubility in comparison to the reference solvent ethanol (EtOH) c(x)/c(EtOH), concentration of curcumin in the respective solvent c(x) (calibration curve in Appendix A), the decadic logarithm of the curcumin concentration log10(c), the calculated chemical potential of curcumin µ(slov), and estimated molar solubility log10(S).

CompoundChemical Structure	Factorc(x)/c(EtOH)	Experimental	COSMO-RS
c(x)(mmol/L)	log10(c)log10 (mol/L)	µ(solv)(kcal/mol)	log10(S)log10 (mol/L)
1.Ethanol 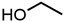	1.00	8.2 ± 0.3	−2.09	−3.26	0.14
2.(*R*)-(+)-Limonene 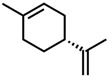	0.05	0.410 ± 0.004	−3.39	3.71	−5.21
3.Citronellol 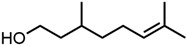	0.25	2.03 ± 0.03	−2.69	−1.65	−1.35
4.1-Octanol 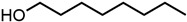	0.25	2.08 ± 0.03	−2.68	−1.97	−1.05
5.Citronellal 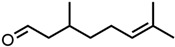	0.75	6.1 ± 0.2	−2.21	−0.55	−2.14
6.trans-2-Hexen-1-ol 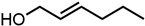	0.81	6.69 ± 0.03	−2.18	−2.48	−0.58
7.Nerol 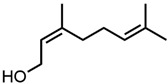	0.82	6.7 ± 0.2	−2.17	−1.75	−1.26
8.trans-Anethole 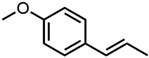	1.09	8.97 ± 0.04	−2.05	0.71	−2.99
9.Triacetin 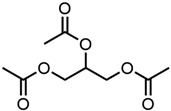	2.36	19.4 ± 0.3	−1.71	−1.95	−1.13
10.Eugenol 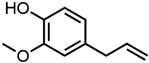	2.82	23.2 ± 0.2	−1.63	−1.34	−1.51
11.α-Ionone 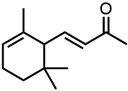	4.65	38 ± 2	−1.42	−0.83	−1.99
12.Vanillin (@17 mol% in ethanol) 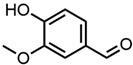	5.85	48.1 ± 0.4 *	−1.32	-	0.00
13.Pseudoionone 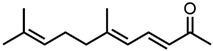	7.13	59 ± 2	−1.23	−1.53	−1.51
14.β-Ionone 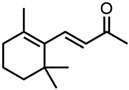	7.15	59 ± 2	−1.23	−1.76	−1.30
15.γ-Dodecalactone 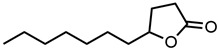	7.62	62.7 ± 0.5	−1.2	−2.07	−1.09
16.Citral 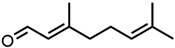	8.26	68.0 ± 0.3	−1.17	−2.69	−0.57
17.Acetone 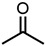	11.38	94 ± 4	−1.03	−3.22	0.08
18.γ-Decalactone 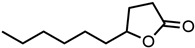	13.17	108 ± 2	−0.96	−2.35	−0.82
19.δ-Tetradecalactone 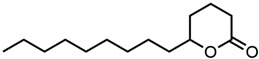	13.19	109 ± 8	−0.96	−1.45	−1.60
20.trans-2-Hexenal 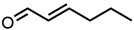	13.31	110 ± 1	−0.96	−3.09	−0.15
21.Benzaldehyde 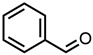	14.47	119 ± 4	−0.92	−2.39	−0.57
22.Methyl-octalactone (Whisky lactone) 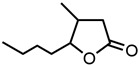	14.68	120.8 ± 0.2	−0.92	−2.68	−0.54
23.(*R*)-(−)-Carvone 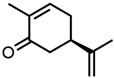	14.82	122 ± 6	−0.91	−1.35	−1.49
24.δ-Dodecalactone 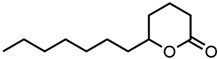	20.84	172 ± 5	−0.77	−2.38	−0.86
25.trans-Cinnamaldehyde 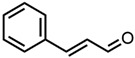	21.00	172.9 ± 0.7	−0.76	−2.91	−0.30
26.δ-Undecalactone 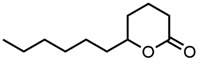	22.03	181 ± 4	−0.74	−2.26	−0.91
27.Massoia lactone 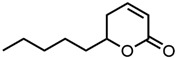	22.27	183 ± 4	−0.74	−2.50	−0.68
28.γ-Octalactone 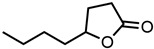	22.76	187 ± 5	−0.73	−2.98	−0.29
29.p-Anisaldehyde 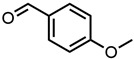	25.55	210 ± 7	−0.68	−3.04	−0.20
30.Sulfolane 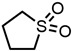	28.20	232 ± 1	−0.63	−3.33	0.08
31.Veratraldehyde (@30 mol% in ethanol) 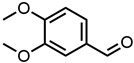	28.64	235.8 ± 0.6 *	−0.63	-	0.00
32.δ-Decalactone 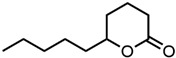	28.94	238 ± 6	−0.62	−2.47	−0.72
33.γ-Caprolactone 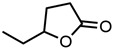	31.95	263 ± 2	−0.58	−3.56	0.15
34.γ-Valerolactone 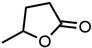	32.59	268 ± 2	−0.57	−3.63	0.23
35.Cyclohexanone 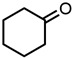	34.71	286 ± 10	−0.54	−2.57	−0.44
36.γ-Butyrolactone 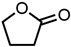	40.81	336 ± 8	−0.47	−3.44	0.18
37.δ-Octalactone 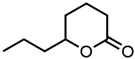	46.24	381 ± 21	−0.42	−3.35	−0.04
38.ε-Caprolactone 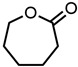	51.18	421 ± 2	−0.38	−4.09	0.40
39.Cyclopentanone 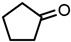	58.22	0479 ± 10	−0.32	−3.10	−0.04
40.δ-Hexalactone 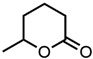	63.48	523 ± 46	−0.28	−4.14	0.42
41.δ-Valerolactone 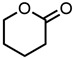	66.10	544 ± 13	−0.26	−4.47	0.52
42.Dimethyl sulfoxide (DMSO) 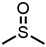	183.34	1510 ± 1	0.18	−5.78	0.00

* Due to vanillin and veratraldehyde being solid aromas, where the curcumin solubility is limited by the solubility of the aroma compound, the highest measurable curcumin concentration was given.

**Table 2 molecules-29-00294-t002:** Calculated ΔG values in (kJ/mol), (kcal/mol), and (kT) of curcumin in different aromas and solvents. The concentration of curcumin in ethanol was used for c_0_.

Compound	Optical Density (a.u.)	c (mol/L)	ΔG (kcal/mol)	ΔG (kT)
trans-Anethole	491	0.009	0.05	0.09
Cinnamyl acetate	1330	0.024	0.64	1.08
Pyruvic acid ^1^	3000	0.055	1.12	1.90
Citral	3723	0.068	1.25	2.11
Acetone	5128	0.094	1.44	2.43
trans-Cinnamaldehyde	9461	0.173	1.80	3.04
γ-Valerolactone	14,684	0.268	2.06	3.48
Cyclopentanone	26,232	0.479	2.41	4.06
δ-Hexalactone	26,690	0.488	2.42	4.08
DMSO	82,603	1.510	3.09	5.21

^1^ Data from Huber et al. [22].

**Table 3 molecules-29-00294-t003:** List of diketo–keto-enol ratios of curcumin taken from ^1^H-NMR spectra (Appendix A). The concentration of curcumin was calculated according to a calibration of curcumin in ethanol (cf. Appendix A). The prepared curcumin samples were diluted with ethanol. Thus, the calibration curve is used to give a rough understanding of the concentration ranges.

Solvent	Diketo–Keto-Enol Ratio	c(Curcumin) (mol/L) in Respective Nondeuterated Solvent
methanol-d4	0.22/2.0 (~10% diketo)	0.008 ^1^
acetone-d6	0.02/2.0 (~1% diketo)	0.09
DMSO-d6	0.02/2.0 (~1% diketo)	1.5

^1^ The solubility of curcumin in methanol was not determined. Instead, the solubility of curcumin in ethanol is given, as it is very similar to the solubility in methanol at 25 °C [42].

**Table 4 molecules-29-00294-t004:** Chemical shifts of curcumin hydrogen atoms (numbering according to Figure 9) in methanol-d4 with 30 mol% additive (citral, δ-hexalactone, and trans-cinnamaldehyde) The respective ^1^H-NMR spectra with assigned signals are shown in the Appendix A. Signals that experienced an upfield shift are marked in bold, and the signals with a downfield shift are shown in italics.

Atom Number	1&4	27	21&24	10&13	11&12	7&16	19&20	3&6	22
methanol-d4	3.91	--	6.63	6.82	7.11	7.22	7.57	--	--
citral	3.91	5.94	6.63	6.83	**7.09**	*7.23*	*7.58*	--	--
δ-hexalactone	*3.93*	6.06	*6.72*	*6.86*	*7.16*	*7.29*	*7.61*	9.07	--
trans-cinnamaldehyde ^1^	**3.76**	--	*6.88*	--	**7.02**	--	*7.60*	--	--

^1^ Data from Huber et al. [11].

**Table 5 molecules-29-00294-t005:** Chemicals used with respective purity, sorted according to manufacturer. The purity data was not available (N/A) for all compounds.

Sigma Aldrich (Darmstadt, Germany)		TCI (Eschborn, Germany)	
Name	Purity	Name	Purity
3,4-Dimethoxybenzaldehyde (Veratraldehyde)	99%	Curcumin (synthetic)	>97.0%
Acetone	≥99.5%	Cyclohexanone	>99.0%
Acetone-d6	99.9 atom%D	Eugenol	>99%
α-Ionone	≥90%	Nerol	>98.0%
β-Ionone	96%	p-Anisaldehyde	>99.0%
Cyclopentanone	>99%	trans-2-hexenal	>97%
Dimethyl sulfoxide (DMSO)	>95%	trans-2-hexenol	>95%
Ethanol	≥99.8%	γ-Butyrolactone	>99.0%
Methanol-d4	99.80%	δ-Hexalactone	>99.0%
Pseudoionone	≥90%	δ-Octalactone	>98.0%
*R*-(+)-Limonene	≥93%	δ-Valerolactone	>98.0%
Sulfolane	99%	ε-Caprolactone	>99.0%
trans-Anethole	≥99%	Fimenich (Satigny, Switzerland)	
γ-Caprolactone	98%	Name	Purity
γ-Decalactone	≥97%	δ-Decalactone	N/A
γ-Dodecalactone	≥97%	δ-Undecalactone	N/A
γ-Octalactone	>97%	δ-Dodecalactone	N/A
γ-Valerolactone	≥98%	δ-Tetradecalactone	N/A
Vanillin	>97%	Citronellol	N/A
		Massoia lactone	NAT
Merck (Darmstadt, Germany)		All Organic Treasures (Wiggensbach, Germany)	
Name	Purity	Name	Purity
Benzaldehyde	99%	Citral	N/A
Citronellal	>96%		
trans-Cinnamaldehyde	>98%		
Deutero (Kastellaun, Germany)		PCW (Parfum Cosmetic World) (Grasse, France)	
Name	Purity	Name	Purity
Hexadeuterodimethyl sulfoxide (DMSO-d6)	99.80%	Methyl-octalactone (Whisky lactone)	98%

## Data Availability

Data are contained within the article and Appendix A.

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
