# Peer review of "Aromas: Lovely to Smell and Nice Solvents for Polyphenols? Curcumin Solubilisation Power of Fragrances and Flavours [Author-notes fn1-molecules-29-00294]"

_molecules, 2024, doi:10.3390/molecules29020294_

Round 1
Reviewer 1 Report
Comments and Suggestions for Authors
This article pays tribute to Professor Farid Chemat, who passed in 2023. It is for the special issue of Molecules untitled "Innovation in Green Extraction and Processing," and its content is perfectly suited to this theme. The authors investigate the solubilizing power of 40 perfumes and flavours towards curcumin, employing COSMO-RS to rationalize the results. This article follows a recent work published by the same group (Sustain. Food Technol. 2023, 1, 319) that already assessed the solubilizing power of Cinnamaldehyde and of some other liquid aromas towards curcumin. In the present manuscript, 40 aromas and fragrances, including over thirty new molecules, are tested individually or in combination with ethanol. The investigation is extensive and of great interest to researchers in the field of natural product extraction. It undoubtedly deserves publication after formal errors are corrected and, more importantly, after the results have been discussed in a more concise yet more in-depth manner by better exploiting the data provided by COSMO-RS.
Minor corrections:
· The text must be reread carefully because I noted 13 typos but there are probably others: (Line N° / Word) 24 collegue, 78 Veratarledehyde, 152 and 611 COMSO-RS, 171 gray, 210 inaccordance, 260 dimethylsulfoxid, 286 refernce, 381 hemiacetale, 383 consideredinvestigated., 394 structre solublity, 396 interactes, 508 biodegardablitly, 607 diekto/keto-enol, 624 motvates
· Line31, Cinnamaldehyde does not possess a phenolic structure.
· The sentence "The simultaneous presence of polyphenols… are the starting point for this study" is allusive. It is necessary to better explain the benefit of extracting curcumin with fragrance molecules, which are significantly more expensive than solvents and probably not as innocuous as one might think. Is it because the extract of fragrance plus curcumin would be directly incorporated into the formulation of end-use products?
· What are the criteria that guided the selection of the 40 molecules tested?
Major Corrections:
· The discussion of the results is difficult to follow because the reader does not see the structural formulas of the 40 molecules, nor their synergistic or antagonistic effects with ethanol.
· It is essential to construct a table showing the names and formulas of the 40 molecules, as well as the calculated (COSMO-RS) and experimental solubilities of curcumin in these pure liquids. Include also in this table the solubility multiplying factor relative to pure ethanol.
· For the COSMO-RS section, show the sigma-profiles and sigma potentials of curcumin and discuss them in terms of relative aptitude for giving or accepting hydrogen bonds.
· In principle, the logarithm of curcumin solubility in the 40 pure solvents (excluding solid compounds) should vary linearly with the chemical potential of curcumin in the solvent. Show the graph illustrating the poor correlation and discuss the discrepancy.
Comments on the Quality of English Languagenumerous typos but the english is correct
Author Response
We sincerely thank the reviewer for their time, in-depth reports, and kind words, which have helped us improve our article. We have modified the article to respond to the report to the best of our ability. Please find below the detailed responses, listed point by point and in the same order as within the report.
Reviewer report:
This article pays tribute to Professor Farid Chemat, who passed in 2023. It is for the special issue of Molecules untitled "Innovation in Green Extraction and Processing," and its content is perfectly suited to this theme. The authors investigate the solubilizing power of 40 perfumes and flavours towards curcumin, employing COSMO-RS to rationalize the results. This article follows a recent work published by the same group (Sustain. Food Technol. 2023, 1, 319) that already assessed the solubilizing power of Cinnamaldehyde and of some other liquid aromas towards curcumin. In the present manuscript, 40 aromas and fragrances, including over thirty new molecules, are tested individually or in combination with ethanol. The investigation is extensive and of great interest to researchers in the field of natural product extraction. It undoubtedly deserves publication after formal errors are corrected and, more importantly, after the results have been discussed in a more concise yet more in-depth manner by better exploiting the data provided by COSMO-RS.
Minor corrections:
The text must be reread carefully because I noted 13 typos but there are probably others: (Line N° / Word) 24 collegue, 78 Veratarledehyde, 152 and 611 COMSO-RS, 171 gray, 210 inaccordance, 260 dimethylsulfoxid, 286 refernce, 381 hemiacetale, 383 consideredinvestigated., 394 structre solublity, 396 interactes, 508 biodegardablitly, 607 diekto/keto-enol, 624 motvates
Line31, Cinnamaldehyde does not possess a phenolic structure.
Response: The script was reread and the writing errors have been cleared.
The sentence "The simultaneous presence of polyphenols… are the starting point for this study" is allusive. It is necessary to better explain the benefit of extracting curcumin with fragrance molecules, which are significantly more expensive than solvents and probably not as innocuous as one might think. Is it because the extract of fragrance plus curcumin would be directly incorporated into the formulation of end-use products?
What are the criteria that guided the selection of the 40 molecules tested?
Response: The introduction was edited accordingly and the selection process for the tested molecules was explained.
Major Corrections:
The discussion of the results is difficult to follow because the reader does not see the structural formulas of the 40 molecules, nor their synergistic or antagonistic effects with ethanol.
It is essential to construct a table showing the names and formulas of the 40 molecules, as well as the calculated (COSMO-RS) and experimental solubilities of curcumin in these pure liquids. Include also in this table the solubility multiplying factor relative to pure ethanol.
Response: While an overview of the molecular structure was shown in the supplementary materials, the proposed table was added to the manuscript and the results of the solubility screening were plotted as heatmaps for a more expressive illustration. The synergistic effects were further discussed in the hydrogen bonding section with respective sigma-profiles.
For the COSMO-RS section, show the sigma-profiles and sigma potentials of curcumin and discuss them in terms of relative aptitude for giving or accepting hydrogen bonds.
Response: The sigma-profiles and sigma-potentials of curcumin and selected molecules were added and discussed to further explain the solubility mechanism of curcumin.
In principle, the logarithm of curcumin solubility in the 40 pure solvents (excluding solid compounds) should vary linearly with the chemical potential of curcumin in the solvent. Show the graph illustrating the poor correlation and discuss the discrepancy.
Response: The proposed figure was added to the solubility screening section and discrepancies were discussed.
Reviewer 2 Report
Comments and Suggestions for Authors
I believe that the manuscript can be accepted in the current form.
Author Response
We sincerely thank the reviewer for their time, and comments, which have helped us improve our article. We have modified the article to respond to the report to the best of our ability. Please find below the detailed responses, listed point by point and in the same order as within the report.
Reviewer report:
I believe that the manuscript can be accepted in the current form.
Response: The abstract, introduction, discussion and conclusion were edited to the referee’s remarks.
Reviewer 3 Report
Comments and Suggestions for Authors
Some minor corrections are needed in the manuscript.

Good
Author Response
We sincerely thank the reviewer for their for their time, in-depth report, and kind words, which have helped us improve our article. We have modified the article to respond to the report to the best of our ability. Please find below the detailed responses, listed point by point and in the same order as within the report.
Reviewer report:
Some minor corrections are needed in the manuscript.
- Abstract is good, but more findings of the research paper can be included and General statement. should be cut short or deleted.
Response: The abstract was edited accordingly.
- . The introduction is written nicely but lack the proper justification of present work, where is methodology adopted with reference to other studied or not should be added.
Response: The introduction was edited accordingly and the proper justification was added.
- In the manuscript material and method section author have give lots of given raw material will purity values better if its was provided in tabular form.
Response: The materials are now provided in tabular form.
- Section 4.21-4.2.4 no single reference from where the methodology have adopted is given therefore add reference.
Response: References were added in the methodology section.
- Line “Apart from classical use in perfumery or as a flavour, new application for trans-cinnamaldehyde and eugenol arise in packaging films due to their antifungal and antibacterial properties which could also be interesting application for mixtures of these aromas with curcumin. The aromatic aroma compounds are generally regarded as safe and have high potential in the solubilisation process of curcumin and other polyphenols”. Needs reference add below reference. Influence of particle size on physical, mechanical, thermal, and morphological properties of tamarind- fenugreek mucilage biodegradable films. Polymer Bulletin .80,3119-3133
Response: While the original draft contains the following reference on this matter, the suggested reference was added as well.
Sanal-Ead, N.; Jangchud, A.; Chonhenchob, V.; Suppakul, P. Antimicrobial Activity of Cinnamaldehyde and Eugenol and Their Activity after Incorporation into Cellulose‐based Packaging Films. Packag. Technol. Sci. 2012, 25, 7–17, doi:10.1002/pts
- Figure.1. Author has given several graphs, please explain the relevant of such complex graphs as these graphs are not in continues form and very difficult to understand from a point of view of readers.
Response: The figures of the solubility screening were changed. A table showing the chemical structures and solubilities is given. Additionally, heatmaps of the solubility screening are provided for clarity.
7.. Statistical analysis is not given by author it should be added in manuscript.
Response: The UV/Vis measurements were performed in triplets and the standard deviation was shown as error bars, cf. Figure 3, 4b, 5, 6. However, for many points, the error bars are not visible as the standard deviation was below 5% for most measurements. The standard deviation of the curcumin concentration in respective pure solvents is listed in Table 1.
- Section 2.1.2,2.1.3 and 2.2.1, are well explained and results are clear to understand
Response: The author thank the referee for their kind words.
- As per the results findings the Conclusion given is bit short, it should be a complete gist of study in bullet points which is easy to understand for the viewers.
Response: The conclusion was edited accordingly and bullet points are provided.
Reviewer 4 Report
Comments and Suggestions for Authors
The authors of the manuscript " Aromas: Lovely to smell and nice solvents for polyphenols? – 2 Curcumin solubilisation power of fragrances and flavours " undertook very interesting research related to the solubility of turmeric in organic compounds. The work is interestingly written, but requires some minor modifications related to the presentation of the results. The authors presented the obtained results in a less expressive way. Due to the large number of results, it would be better to visualize the results graphically using heat maps. Additionally, I suggest including a table in the text with results showing the solubility of turmeric in the solvents used.
Author Response
We sincerely thank the reviewer for their for their time, in-depth reports, and kind words, which have helped us improve our article. We have modified the article to respond to the report to the best of our ability. Please find below the detailed responses, listed point by point and in the same order as within the report.
Reviewer report:
The authors of the manuscript " Aromas: Lovely to smell and nice solvents for polyphenols? – 2 Curcumin solubilisation power of fragrances and flavours " undertook very interesting research related to the solubility of turmeric in organic compounds. The work is interestingly written, but requires some minor modifications related to the presentation of the results. The authors presented the obtained results in a less expressive way. Due to the large number of results, it would be better to visualize the results graphically using heat maps. Additionally, I suggest including a table in the text with results showing the solubility of turmeric in the solvents used.
Response: A table showing the chemical structures and solubilities and heatmaps are provided for better visualization of the results.